# Polynomial Codes: an Optimal Design for High-Dimensional Coded Matrix Multiplication

**Qian Yu\*, Mohammad Ali Maddah-Ali†, and A. Salman Avestimehr\***
\* Department of Electrical Engineering, University of Southern California, Los Angeles, CA, USA
† Nokia Bell Labs, Holmdel, NJ, USA

## Abstract

We consider a large-scale matrix multiplication problem where the computation is carried out using a distributed system with a master node and multiple worker nodes, where each worker can store parts of the input matrices. We propose a computation strategy that leverages ideas from coding theory to design intermediate computations at the worker nodes, in order to optimally deal with straggling workers. The proposed strategy, named as *polynomial codes*, achieves the optimum recovery threshold, defined as the minimum number of workers that the master needs to wait for in order to compute the output. This is the first code that achieves the optimal utilization of redundancy for tolerating stragglers or failures in distributed matrix multiplication. Furthermore, by leveraging the algebraic structure of polynomial codes, we can map the reconstruction problem of the final output to a polynomial interpolation problem, which can be solved efficiently. Polynomial codes provide order-wise improvement over the state of the art in terms of recovery threshold, and are also optimal in terms of several other metrics including computation latency and communication load. Moreover, we extend this code to distributed convolution and show its order-wise optimality.

## 1 Introduction

Matrix multiplication is one of the key building blocks underlying many data analytics and machine learning algorithms. Many such applications require massive computation and storage power to process large-scale datasets. As a result, distributed computing frameworks such as Hadoop MapReduce [1] and Spark [2] have gained significant traction, as they enable processing of data sizes at the order of tens of terabytes and more.

As we scale out computations across many distributed nodes, a major performance bottleneck is the latency in waiting for slowest nodes, or "stragglers" to finish their tasks [3]. The current approaches to mitigate the impact of stragglers involve creation of some form of "computation redundancy". For example, *replicating* the straggling task on another available node is a common approach to deal with stragglers (e.g., [4]). However, there have been recent results demonstrating that *coding* can play a transformational role for creating and exploiting computation redundancy to effectively alleviate the impact of stragglers [5, 6, 7, 8, 9]. Our main result in this paper is the development of optimal codes, named *polynomial codes*, to deal with stragglers in distributed high-dimensional matrix multiplication, which also provides *order-wise* improvement over the state of the art.

More specifically, we consider a distributed matrix multiplication problem where we aim to compute $C = A^\intercal B$ from input matrices $A$ and $B$. As shown in Fig. 1, the computation is carried out using a distributed system with a master node and $N$ worker nodes that can each store $\frac{1}{m}$ fraction of $A$ and $\frac{1}{n}$ fraction of $B$, for some parameters $m, n \in \mathbb{N}^+$. We denote the stored submtarices at each

worker $i \in \{0, \ldots, N-1\}$ by $\tilde{A}_i$ and $\tilde{B}_i$, which can be designed as arbitrary functions of $A$ and $B$ respectively. Each worker $i$ then computes the product $\tilde{A}_i^\mathsf{T}\tilde{B}_i$ and returns the result to the master.

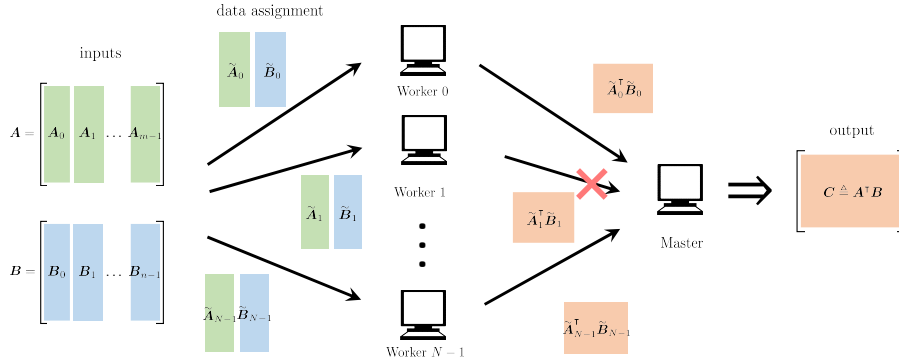

Figure 1: Overview of the distributed matrix multiplication framework. Coded data are initially stored distributedly at $N$ workers according to data assignment. Each worker computes the product of the two stored matrices and returns it to the master. By carefully designing the computation strategy, the master can decode given the computing results from a subset of workers, without having to wait for the stragglers (worker 1 in this example).

By carefully designing the computation strategy at each worker (i.e. designing $\tilde{A}_i$ and $\tilde{B}_i$), the master only needs to wait for the fastest subset of workers before recovering output $C$, hence mitigating the impact of stragglers. Given a computation strategy, we define its *recovery threshold* as the minimum number of workers that the master needs to wait for in order to compute $C$. In other words, if *any* subset of the workers with size no smaller than the *recovery threshold* finish their jobs, the master is able to compute $C$. Given this formulation, we are interested in the following main problem.

> What is the minimum possible recovery threshold for distributed matrix multiplication? Can we find an optimal computation strategy that achieves the minimum recovery threshold, while allowing efficient decoding of the final output at the master node?

There have been two computing schemes proposed earlier for this problem that leverage ideas from coding theory. The first one, introduced in [5] and extended in [10], injects redundancy in only one of the input matrices using maximum distance separable (MDS) codes [11] [1]. We illustrate this approach, referred to as *one dimensional MDS code* (*1D MDS code*), using the example shown in Fig. 2a, where we aim to compute $C = A^\mathsf{T}B$ using 3 workers that can each store half of $A$ and the entire $B$. The 1D MDS code evenly divides $A$ along the column into two submatrices denoted by $A_0$ and $A_1$, encodes them into 3 coded matrices $A_0$, $A_1$, and $A_0 + A_1$, and then assigns them to the 3 workers. This design allows the master to recover the final output given the results from any 2 of the 3 workers, hence achieving a recovery threshold of 2. More generally, one can show that the 1D MDS code achieves a recovery threshold of

$$K_{\text{1D-MDS}} \triangleq N - \frac{N}{n} + m = \Theta(N). \tag{1}$$

An alternative computing scheme was recently proposed in [10] for the case of $m = n$, referred to as the *product code*, which instead injects redundancy in both input matrices. This coding technique has also been proposed earlier in the context of Fault Tolerant Computing in [12, 13]. As demonstrated in Fig. 2b, product code aligns workers in an $\sqrt{N}-$by$-\sqrt{N}$ layout. $A$ is divided along the columns into $m$ submatrices, encoded using an $(\sqrt{N}, m)$ MDS code into $\sqrt{N}$ coded matrices, and then assigned to the $\sqrt{N}$ columns of workers. Similarly $\sqrt{N}$ coded matrices of $B$ are created and assigned to the $\sqrt{N}$ rows. Given the property of MDS codes, the master can decode an entire row after obtaining any $m$ results in that row; likewise for the columns. Consequently, the master can recover the final output using a peeling algorithm, iteratively decoding the MDS codes on rows and columns until the output $C$ is completely available. For example, if the 5 computing results $A_1^\mathsf{T}B_0$, $A_1^\mathsf{T}B_1$, $(A_0 + A_1)^\mathsf{T}B_1$, $A_0^\mathsf{T}(B_0 + B_1)$, and $A_1^\mathsf{T}(B_0 + B_1)$ are received as demonstrated in Fig. 2b, the master can recover the

needed results by computing $A_0^\mathsf{T} B_1 = (A_0 + A_1)^\mathsf{T} B_1 - A_1^\mathsf{T} B_1$ then $A_0^\mathsf{T} B_0 = A_0^\mathsf{T}(B_0 + B_1) - A_0^\mathsf{T} B_1$. In general, one can show that the product code achieves a recovery threshold of

$$K_{\text{product}} \triangleq 2(m-1)\sqrt{N} - (m-1)^2 + 1 = \Theta(\sqrt{N}), \tag{2}$$

which significantly improves over $K_{\text{1D-MDS}}$.

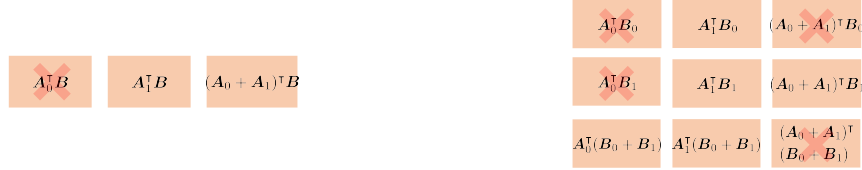

(a) 1D MDS-code [5] in an example with 3 workers that can each store half of $A$ and the entire $B$.

(b) Product code [10] in an example with 9 workers that can each store half of $A$ and half of $B$.

Figure 2: Illustration of (a) 1D MDS code, and (b) product code.

In this paper, we show that quite interestingly, the optimum recovery threshold can be far less than what the above two schemes achieve. In fact, we show that the minimum recovery threshold does not scale with the number of workers (i.e. $\Theta(1)$). We prove this fact by designing a novel coded computing strategy, referred to as the *polynomial code*, which achieves the optimum recovery threshold of $mn$, and significantly improves the state of the art. Hence, our main result is as follows.

> For a general matrix multiplication task $C = A^\mathsf{T} B$ using $N$ workers, where each worker can store $\frac{1}{m}$ fraction of $A$ and $\frac{1}{n}$ fraction of $B$, we propose *polynomial codes* that achieve the *optimum* recovery threshold of
>
> $$K_{\text{poly}} \triangleq mn = \Theta(1). \tag{3}$$
>
> Furthermore, polynomial code only requires a decoding complexity that is almost linear to the input size.

The main novelty and advantage of the proposed polynomial code is that, by carefully designing the algebraic structure of the encoded submatrices, we ensure that *any* $mn$ intermediate computations at the workers are sufficient for recovering the final matrix multiplication product at the master. This in a sense creates an MDS structure on the intermediate computations, instead of only the encoded matrices as in prior works. Furthermore, by leveraging the algebraic structure of polynomial codes, we can then map the reconstruction problem of the final output at the master to a polynomial interpolation problem (or equivalently Reed-Solomon decoding [14]), which can be solved efficiently [15]. This mapping also bridges the rich theory of algebraic coding and distributed matrix multiplication.

We prove the optimality of polynomial code by showing that it achieves the information theoretic lower bound on the recovery threshold, obtained by cut-set arguments (i.e., we need at least $mn$ matrix blocks returned from workers to recover the final output, which exactly have size $mn$ blocks). Hence, the proposed polynomial code essentially enables a specific computing strategy such that, from any subset of workers that give the minimum amount of information needed to recover $C$, the master can successfully decode the final output. As a by-product, we also prove the optimality of polynomial code under several other performance metrics considered in previous literature: computation latency [5, 10], probability of failure given a deadline [9], and communication load [16, 17, 18].

We extend the polynomial code to the problem of distributed convolution [9]. We show that by simply reducing the convolution problem to matrix multiplication and applying the polynomial code, we strictly and unboundedly improve the state of the art. Furthermore, by exploiting the computing structure of convolution, we propose a variation of the polynomial code, which strictly reduces the recovery threshold even further, and achieves the optimum recovery threshold within a factor of 2.

Finally, we implement and benchmark the polynomial code on an Amazon EC2 cluster. We measure the computation latency and empirically demonstrate its performance gain under straggler effects.

## 2 System Model, Problem Formulation, and Main Result

We consider a problem of matrix multiplication with two input matrices $A \in \mathbb{F}_q^{s \times r}$ and $B \in \mathbb{F}_q^{s \times t}$, for some integers $r$, $s$, $t$ and a sufficiently large finite field $\mathbb{F}_q$. We are interested in computing the product $C \triangleq A^\intercal B$ in a distributed computing environment with a master node and $N$ worker nodes, where each worker can store $\frac{1}{m}$ fraction of $A$ and $\frac{1}{n}$ fraction of $B$, for some parameters $m, n \in \mathbb{N}^+$ (see Fig. 1). We assume at least one of the two input matrices $A$ and $B$ is tall (i.e. $s \geq r$ or $s \geq t$), because otherwise the output matrix $C$ would be rank inefficient and the problem is degenerated.

Specifically, each worker $i$ can store two matrices $\tilde{A}_i \in \mathbb{F}_q^{s \times \frac{r}{m}}$ and $\tilde{B}_i \in \mathbb{F}_q^{s \times \frac{t}{n}}$, computed based on *arbitrary functions* of $A$ and $B$ respectively. Each worker can compute the product $\tilde{C}_i \triangleq \tilde{A}_i^\intercal \tilde{B}_i$, and return it to the master. The master waits only for the results from a subset of workers, before proceeding to recover the final output $C$ given these products using certain *decoding functions*.[2]

### 2.1 Problem Formulation

Given the above system model, we formulate the *distributed matrix multiplication problem* based on the following terminology: We define the *computation strategy* as the $2N$ functions, denoted by

$$\boldsymbol{f} = (f_0, f_1, ..., f_{N-1}), \qquad \boldsymbol{g} = (g_0, g_1, ..., g_{N-1}), \tag{4}$$

that are used to compute each $\tilde{A}_i$ and $\tilde{B}_i$. Specifically,

$$\tilde{A}_i = f_i(A), \qquad \tilde{B}_i = g_i(B), \qquad \forall\, i \in \{0, 1, ..., N-1\}. \tag{5}$$

For any integer $k$, we say a computation strategy is *k-recoverable* if the master can recover $C$ given the computing results from *any* $k$ workers. We define the *recovery threshold* of a computation strategy, denoted by $k(\boldsymbol{f}, \boldsymbol{g})$, as the minimum integer $k$ such that computation strategy $(\boldsymbol{f}, \boldsymbol{g})$ is $k$-recoverable.

Using the above terminology, we define the following concept:

**Definition 1.** For a distributed matrix multiplication problem of computing $A^\intercal B$ using $N$ workers that can each store $\frac{1}{m}$ fraction of $A$ and $\frac{1}{n}$ fraction of $B$, we define the *optimum recovery threshold*, denoted by $K^*$, as the minimum achievable recovery threshold among all computation strategies, i.e.

$$K^* \triangleq \min_{\boldsymbol{f}, \boldsymbol{g}} k(\boldsymbol{f}, \boldsymbol{g}). \tag{6}$$

The goal of this problem is to find the optimum recovery threshold $K^*$, as well as a computation strategy that achieves such an optimum threshold.

### 2.2 Main Result

Our main result is stated in the following theorem:

**Theorem 1.** *For a distributed matrix multiplication problem of computing $A^\intercal B$ using $N$ workers that can each store $\frac{1}{m}$ fraction of $A$ and $\frac{1}{n}$ fraction of $B$, the minimum recovery threshold $K^*$ is*

$$K^* = mn. \tag{7}$$

*Furthermore, there is a computation strategy, referred to as the* polynomial code, *that achieves the above $K^*$ while allowing efficient decoding at the master node, i.e., with complexity equal to that of polynomial interpolation given $mn$ points.*

*Remark* 1. Compared to the state of the art [5, 10], the polynomial code provides order-wise improvement in terms of the recovery threshold. Specifically, the recovery thresholds achieved by 1D MDS code [5, 10] and product code [10] scale linearly with $N$ and $\sqrt{N}$ respectively, while the proposed polynomial code actually achieves a recovery threshold that does not scale with $N$. Furthermore, polynomial code achieves the optimal recovery threshold. To the best of our knowledge, this is the first optimal design proposed for the distributed matrix multiplication problem.

*Remark* 2. We prove the optimality of polynomial code using a matching information theoretic lower bound, which is obtained by applying a cut-set type argument around the master node. As a by-product, we can also prove that the polynomial code simultaneously achieves optimality in terms of several other performance metrics, including the computation latency [5, 10], the probability of failure given a deadline [9], and the communication load [16, 17, 18], as discussed in Section 3.4.

*Remark* 3. The polynomial code not only improves the state of the art asymptotically, but also gives strict and significant improvement for any parameter values of $N$, $m$, and $n$ (See Fig. 3 for example).

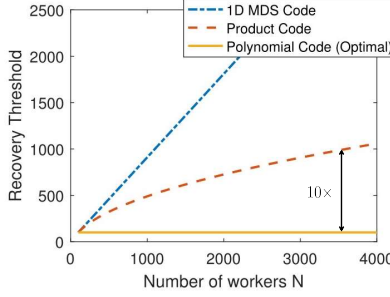

Figure 3: Comparison of the recovery thresholds achieved by the proposed polynomial code and the state of the arts (1D MDS code [5] and product code [10]), where each worker can store $\frac{1}{10}$ fraction of each input matrix. The polynomial code attains the optimum recovery threshold $K^*$, and significantly improves the state of the art.

*Remark* 4. As we will discuss in Section 3.2, decoding polynomial code can be mapped to a polynomial interpolation problem, which can be solved in time almost linear to the input size [15]. This is enabled by carefully designing the computing strategies at the workers, such that the computed products form a *Reed-Solomon code* [19] , which can be decoded efficiently using any polynomial interpolation algorithm or Reed-Solomon decoding algorithm that provides the best performance depending on the problem scenario (e.g., [20]).

*Remark* 5. Polynomial code can be extended to other distributed computation applications involving linear algebraic operations. In Section 4, we focus on the problem of distributed convolution, and show that we can obtain order-wise improvement over the state of the art (see [9]) by directly applying the polynomial code. Furthermore, by exploiting the computing structure of convolution, we propose a variation of the polynomial code that achieves the optimum recovery threshold within a factor of 2.

*Remark* 6. In this work we focused on designing optimal coding techniques to handle stragglers issues. The same technique can also be applied to the fault tolerance computing setting (e.g., within the algorithmic fault tolerance computing framework of [12, 13], where a module can produce arbitrary error results under failure), to improve robustness to failures in computing. Given that the polynomial code produces computing results that are coded by *Reed-Solomon code*, which has the optimum hamming distance, it allows detecting, or correcting the maximum possible number of module errors. Specifically, polynomial code can robustly detect up to $N - mn$ errors, and correct up to $\lfloor \frac{N-mn}{2} \rfloor$ errors. This provides the first optimum code for matrix multiplication under fault tolerance computing.

## 3 Polynomial Code and Its Optimality

In this section, we formally describe the polynomial code and its decoding procedure. We then prove its optimality with an information theoretic converse, which completes the proof of Theorem 1. Finally, we conclude this section with the optimality of polynomial code under other settings.

### 3.1 Motivating Example

We first demonstrate the main idea through a motivating example. Consider a distributed matrix multiplication task of computing $C = A^\intercal B$ using $N = 5$ workers that can each store half of the matrices (see Fig. 4). We evenly divide each input matrix along the column side into 2 submatrices:

$$A = [A_0 \ A_1], \qquad B = [B_0 \ B_1]. \tag{8}$$

Given this notation, we essentially want to compute the following 4 uncoded components:

$$C = A^\intercal B = \begin{bmatrix} A_0^\intercal B_0 & A_0^\intercal B_1 \\ A_1^\intercal B_0 & A_1^\intercal B_1 \end{bmatrix}. \tag{9}$$

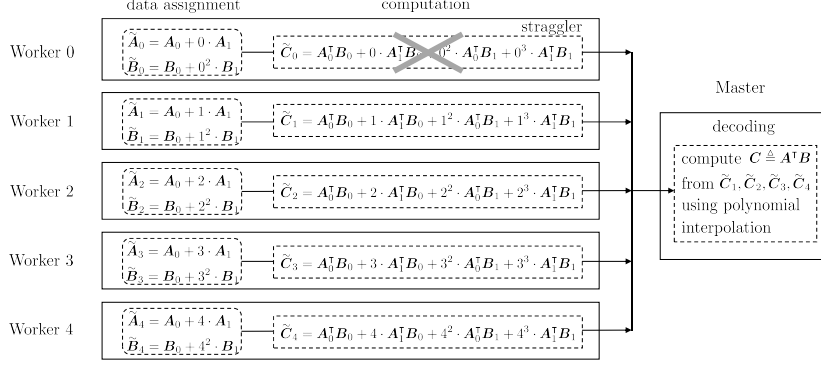

Figure 4: Example using polynomial code, with 5 workers that can each store half of each input matrix. (a) Computation strategy: each worker $i$ stores $A_0 + iA_1$ and $B_0 + i^2 B_1$, and computes their product. (b) Decoding: master waits for results from *any* 4 workers, and decodes the output using fast polynomial interpolation algorithm.

Now we design a computation strategy to achieve the optimum recovery threshold of 4. Suppose elements of $A, B$ are in $\mathbb{F}_7$, let each worker $i \in \{0, 1, ..., 4\}$ store the following two coded submatrices:

$$\tilde{A}_i = A_0 + iA_1, \qquad \tilde{B}_i = B_0 + i^2 B_1. \tag{10}$$

To prove that this design gives a recovery threshold of 4, we need to design a valid decoding function for any subset of 4 workers. We demonstrate this decodability through a representative scenario, where the master receives the computation results from workers 1, 2, 3, and 4, as shown in Figure 4. The decodability for the other 4 possible scenarios can be proved similarly.

According to the designed computation strategy, we have

$$\begin{bmatrix} \tilde{C}_1 \\ \tilde{C}_2 \\ \tilde{C}_3 \\ \tilde{C}_4 \end{bmatrix} = \begin{bmatrix} 1^0 & 1^1 & 1^2 & 1^3 \\ 2^0 & 2^1 & 2^2 & 2^3 \\ 3^0 & 3^1 & 3^2 & 3^3 \\ 4^0 & 4^1 & 4^2 & 4^3 \end{bmatrix} \begin{bmatrix} A_0^\intercal B_0 \\ A_1^\intercal B_0 \\ A_0^\intercal B_1 \\ A_1^\intercal B_1 \end{bmatrix}. \tag{11}$$

The coefficient matrix in the above equation is a Vandermonde matrix, which is invertible because its parameters $1, 2, 3, 4$ are distinct in $\mathbb{F}_7$. So one way to recover $C$ is to directly invert equation (11), which proves the decodability. However, directly computing this inverse using the classical inversion algorithm might be expensive in more general cases. Quite interestingly, because of the algebraic structure we designed for the computation strategy (i.e., equation (10)), the decoding process can be viewed as a polynomial interpolation problem (or equivalently, decoding a Reed-Solomon code).

Specifically, in this example each worker $i$ returns

$$\tilde{C}_i = \tilde{A}_i^\intercal \tilde{B}_i = A_0^\intercal B_0 + iA_1^\intercal B_0 + i^2 A_0^\intercal B_1 + i^3 A_1^\intercal B_1, \tag{12}$$

which is essentially the value of the following polynomial at point $x = i$:

$$h(x) \triangleq A_0^\intercal B_0 + xA_1^\intercal B_0 + x^2 A_0^\intercal B_1 + x^3 A_1^\intercal B_1. \tag{13}$$

Hence, recovering $C$ using computation results from 4 workers is equivalent to interpolating a 3rd-degree polynomial given its values at 4 points. Later in this section, we will show that by mapping the decoding process to polynomial interpolation, we can achieve almost-linear decoding complexity.

## 3.2 General Polynomial Code

Now we present the polynomial code in a general setting that achieves the optimum recovery threshold stated in Theorem 1 for any parameter values of $N$, $m$, and $n$. First of all, we evenly divide each input matrix along the column side into $m$ and $n$ submatrices respectively, i.e.,

$$A = [A_0 \ A_1 \ ... \ A_{m-1}], \qquad B = [B_0 \ B_1 \ ... \ B_{n-1}], \tag{14}$$

We then assign each worker $i \in \{0, 1, ..., N-1\}$ a number in $\mathbb{F}_q$, denoted by $x_i$, and make sure that all $x_i$'s are distinct. Under this setting, we define the following class of computation strategies.

**Definition 2.** Given parameters $\alpha, \beta \in \mathbb{N}$, we define the $(\alpha, \beta)$-polynomial code as

$$\tilde{A}_i = \sum_{j=0}^{m-1} A_j x_i^{j\alpha}, \qquad \tilde{B}_i = \sum_{j=0}^{n-1} B_j x_i^{j\beta}, \qquad \forall\, i \in \{0, 1, ..., N-1\}. \tag{15}$$

In an $(\alpha, \beta)$-polynomial code, each worker $i$ essentially computes

$$\tilde{C}_i = \tilde{A}_i^\intercal \tilde{B}_i = \sum_{j=0}^{m-1} \sum_{k=0}^{n-1} A_j^\intercal B_k x_i^{j\alpha + k\beta}. \tag{16}$$

In order for the master to recover the output given any $mn$ results (i.e. achieve the optimum recovery threshold), we carefully select the design parameters $\alpha$ and $\beta$, while making sure that no two terms in the above formula has the same exponent of $x$. One such choice is $(\alpha, \beta) = (1, m)$, i.e, let

$$\tilde{A}_i = \sum_{j=0}^{m-1} A_j x_i^j, \qquad \tilde{B}_i = \sum_{j=0}^{n-1} B_j x_i^{jm}. \tag{17}$$

Hence, each worker computes the value of the following degree $mn - 1$ polynomial at point $x = x_i$:

$$h(x) \triangleq \sum_{j=0}^{m-1} \sum_{k=0}^{n-1} A_j^\intercal B_k x^{j+km}, \tag{18}$$

where the coefficients are exactly the $mn$ uncoded components of $C$. Since all $x_i$'s are selected to be distinct, recovering $C$ given results from any $mn$ workers is essentially interpolating $h(x)$ using $mn$ distinct points. Since $h(x)$ has degree $mn - 1$, the output $C$ can always be uniquely decoded.

In terms of complexity, this decoding process can be viewed as interpolating degree $mn - 1$ polynomials of $\mathbb{F}_q$ for $\frac{rt}{mn}$ times. It is well known that polynomial interpolation of degree $k$ has a complexity of $O(k \log^2 k \log \log k)$ [15]. Therefore, decoding polynomial code also only requires a complexity of $O(rt \log^2(mn) \log \log(mn))$. Furthermore, this complexity can be reduced by simply swapping in any faster polynomial interpolation algorithm or Reed-Solomon decoding algorithm.

*Remark* 7. We can naturally extend polynomial code to the scenario where input matrix elements are real or complex numbers. In practical implementation, to avoid handling large elements in the coefficient matrix, we can first quantize input values into numbers of finite digits, embed them into a finite field that covers the range of possible values of the output matrix elements, and then directly apply polynomial code. By embedding into finite fields, we avoid large intermediate computing results, which effectively saves storage and computation time, and reduces numerical errors.

## 3.3 Optimality of Polynomial Code for Recovery Threshold

So far we have constructed a computing scheme that achieves a recovery threshold of $mn$, which upper bounds $K^*$. To complete the proof of Theorem 1, here we establish a matching lower bound through an information theoretic converse.

We need to prove that for any computation strategy, the master needs to wait for at least $mn$ workers in order to recover the output. Recall that at least one of $A$ and $B$ is a tall matrix. Without loss of generality, assume $A$ is tall (i.e. $s \geq r$). Let $A$ be an arbitrary fixed full rank matrix and $B$ be sampled from $\mathbb{F}_q^{s \times t}$ uniformly at random. It is easy to show that $C = A^\intercal B$ is uniformly distributed on $\mathbb{F}_q^{r \times t}$. This means that the master essentially needs to recover a random variable with entropy of $H(C) = rt \log_2 q$ bits. Note that each worker returns $\frac{rt}{mn}$ elements of $\mathbb{F}_q$, providing at most $\frac{rt}{mn} \log_2 q$ bits of information. Consequently, using a cut-set bound around the master, we can show that at least $mn$ results from the workers need to be collected, and thus we have $K^* \geq mn$.

*Remark* 8 (Random Linear Code). We conclude this subsection by noting that, another computation design is to let each worker store two random linear combinations of the input submatrices. Although this design can achieve the optimal recovery threshold with high probability, it creates a large coding overhead and requires high decoding complexity (e.g., $O(m^3 n^3 + mnrt)$ using the classical inversion decoding algorithm). Compared to random linear code, the proposed polynomial code achieves the optimum recovery threshold deterministically, with a significantly lower decoding complexity.

### 3.4 Optimality of Polynomial Code for Other Performance Metrics

In the previous subsection, we proved that polynomial code is optimal in terms of the recovery threshold. As a by-product, we can prove that it is also optimal in terms of some other performance metrics. In particular, we consider the following 3 metrics considered in prior works, and formally establish the optimality of polynomial code for each of them. Proofs can be found in Appendix A.

**Computation latency** is considered in models where the computation time $T_i$ of each worker $i$ is a random variable with a certain probability distribution (e.g, [5, 10]). The computation latency is defined as the amount of time required for the master to collect enough information to decode $C$.

**Theorem 2.** *For any computation strategy, the computation latency $T$ is always no less than the latency achieved by polynomial code, denoted by $T_{\text{poly}}$. Namely,*

$$T \geq T_{\text{poly}}. \tag{19}$$

**Probability of failure given a deadline** is defined as the probability that the master does not receive enough information to decode $C$ at any time $t$ [9].

**Corollary 1.** *For any computation strategy, let $T$ denote its computation latency, and let $T_{\text{poly}}$ denote the computation latency of polynomial code. We have*

$$\mathbb{P}(T > t) \geq \mathbb{P}(T_{\text{poly}} > t) \quad \forall\, t \geq 0. \tag{20}$$

Corollary 1 directly follows from Theorem 2 since (19) implies (20) .

**Communication load** is another important metric in distributed computing (e.g. [16, 17, 18]), defined as the minimum number of bits needed to be communicated in order to complete the computation.

**Theorem 3.** *Polynomial code achieves the minimum communication load for distributed matrix multiplication, which is given by*

$$L^* = rt \log_2 q. \tag{21}$$

## 4 Extension to Distributed Convolution

We can extend our proposed polynomial code to distributed convolution. Specifically, we consider a convolution task with two input vectors

$$\boldsymbol{a} = [\boldsymbol{a}_0\ \boldsymbol{a}_1\ ...\ \boldsymbol{a}_{m-1}], \qquad \boldsymbol{b} = [\boldsymbol{b}_0\ \boldsymbol{b}_1\ ...\ \boldsymbol{b}_{n-1}], \tag{22}$$

where all $\boldsymbol{a}_i$'s and $\boldsymbol{b}_i$'s are vectors of length $s$ over a sufficiently large field $\mathbb{F}_q$. We want to compute $\boldsymbol{c} \triangleq \boldsymbol{a} * \boldsymbol{b}$ using a master and $N$ workers. Each worker can store two vectors of length $s$, which are functions of $\boldsymbol{a}$ and $\boldsymbol{b}$ respectively. We refer to these functions as the *computation strategy*.

Each worker computes the convolution of its stored vectors, and returns it to the master. The master only waits for the fastest subset of workers, before proceeding to decode $\boldsymbol{c}$. Similar to distributed matrix multiplication, we define the *recovery threshold* for each computation strategy. We aim to characterize the optimum recovery threshold denoted by $K_{\text{conv}}^*$, and find computation strategies that closely achieve this optimum threshold, while allowing efficient decoding at the master.

Distributed convolution has also been studied in [9], where the *coded convolution scheme* was proposed. The main idea of the coded convolution scheme is to inject redundancy in only one of the input vectors using MDS codes. The master waits for enough results such that all intermediate values $\boldsymbol{a}_i * \boldsymbol{b}_j$ can be recovered, which allows the final output to be computed. One can show that this coded convolution scheme is in fact equivalent to the 1D MDS-coded scheme proposed in [10]. Consequently, it achieves a recovery threshold of $K_{\text{1D-MDS}} = N - \frac{N}{n} + m$.

Note that by simply adapting our proposed polynomial code designed for distributed matrix multiplication to distributed convolution, the master can recover all intermediate values $\boldsymbol{a}_i * \boldsymbol{b}_j$ after receiving results from *any $mn$* workers, to decode the final output. Consequently, this achieves a recovery threshold of $K_{\text{poly}} = mn$, which already strictly and significantly improves the state of the art.

In this paper, we take one step further and propose an improved computation strategy, strictly reducing the recovery threshold on top of the naive polynomial code. The result is summarized as follows:

**Theorem 4.** *For a distributed convolution problem of computing $\boldsymbol{a} * \boldsymbol{b}$ using $N$ workers that can each store $\frac{1}{m}$ fraction of $\boldsymbol{a}$ and $\frac{1}{n}$ fraction of $\boldsymbol{b}$, we can find a computation strategy that achieves a recovery threshold of*

$$K_{\text{conv-poly}} \triangleq m + n - 1. \tag{23}$$

*Furthermore, this computation strategy allows efficient decoding, i.e., with complexity equal to that of polynomial interpolation given $m + n - 1$ points.*

We prove Theorem 4 by proposing a variation of the polynomial code, which exploits the computation structure of convolution. This new computing scheme is formally demonstrated in Appendix B.

*Remark* 9. Similar to distributed matrix multiplication, our proposed computation strategy provides orderwise improvement compared to state of the art [9] in various settings. Furthermore, it achieves almost-linear decoding complexity using the fastest polynomial interpolation algorithm or the Reed-Solomon decoding algorithm. More recently, we have shown that this proposed scheme achieves the optimum recovery threshold among all computation strategies that are linear functions [21].

Moreover, we characterize $K_{\text{conv}}$ within a factor of 2, as stated in the following theorem and proved in Appendix C.

**Theorem 5.** *For a distributed convolution problem, the minimum recovery threshold $K^*_{\text{conv}}$ can be characterized within a factor of* 2*, i.e.:*

$$\frac{1}{2}K_{\text{conv-poly}} < K^*_{\text{conv}} \leq K_{\text{conv-poly}}. \tag{24}$$

## 5  Experiment Results

To examine the efficiency of our proposed polynomial code, we implement the algorithm in Python using the mpi4py library and deploy it on an AWS EC2 cluster of $18$ nodes, with the master running on a c1.medium instance, and $17$ workers running on m1.small instances.

The input matrices are randomly generated as two `numpy` matrices of size $4000$ by $4000$, and then encoded and assigned to the workers in the preprocessing stage. Each worker stores $\frac{1}{4}$ fraction of each input matrix. In the computation stage, each worker computes the product of their assigned matrices, and then returns the result using `MPI.Comm.Isend()`. The master actively listens to responses from the 17 worker nodes through `MPI.Comm.Irecv()`, and uses `MPI.Request.Waitany()` to keep polling for the earliest fulfilled request. Upon receiving 16 responses, the master stops listening and starts decoding the result. To achieve the best performance, we implement an FFT-based algorithm for the Reed-Solomon decoding.

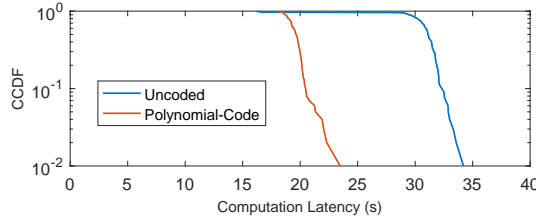

Figure 5: Comparison of polynomial code and the uncoded scheme. We implement polynomial code and the uncoded scheme using Python and mpi4py library and deploy them on an Amazon EC2 cluster of 18 instances. We measure the computation latency of both algorithms and plot their CCDF. Polynomial code can reduce the tail latency by 37% even taking into account of the decoding overhead.

We compare our results with distributed matrix multiplication without coding.[3] The uncoded implementation is similar, except that only 16 out of the 17 workers participate in the computation, each of them storing and processing $\frac{1}{4}$ fraction of uncoded rows from each input matrix. The master waits for all 16 workers to return, and does not need to perform any decoding algorithm to recover the result.

To simulate straggler effects in large-scale systems, we compare the computation latency of these two schemes in a setting where a randomly picked worker is running a background thread which approximately doubles the computation time. As shown in Fig. 5, polynomial code can reduce the tail latency by 37% in this setting, even taking into account of the decoding overhead.

## 6  Acknowledgement

This work is in part supported by NSF grants CCF-1408639, NETS-1419632, ONR award N000141612189, NSA grant, and a research gift from Intel. This material is based upon work supported by Defense Advanced Research Projects Agency (DARPA) under Contract No. HR001117C0053. The views, opinions, and/or findings expressed are those of the author(s) and should not be interpreted as representing the official views or policies of the Department of Defense or the U.S. Government.

## Footnotes

[1]An $(n, k)$ MDS code is a linear code which transforms $k$ raw inputs to $n$ coded outputs, such that from any subset of size $k$ of the outputs, the original $k$ inputs can be recovered.

[2]Note that we consider the most general model and do not impose any constraints on the decoding functions. However, any good decoding function should have relatively low computation complexity.

[3]Due to the EC2 instance request quota limit of 20, 1D MDS code and product code could not be implemented in this setting, which require at least 21 and 26 nodes respectively.

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
