[Supplementary Material]

# Appendix A  Optimality of Polynomial Code in Latency and Communication Load

In this section we prove the optimality of polynomial code for distributed matrix multiplication in terms of computation latency and communication load. Specifically, we provide the proof of Theorem 2 and Theorem 3.

## A.1  Proof of Theorem 2

Consider an arbitrary computation strategy, we denote its computation latency by $T$. By definition, $T$ is given as follows:

$$T = \min\{\, t \geq 0 \mid C \text{ is decodable given results from all workers in } \{\, i \mid T_i \leq t \,\} \,\}, \qquad (1)$$

where $T_i$ denotes the computation time of worker $i$. To simplify the discussion, we define

$$\mathcal{S}(t) = \{\, i \mid T_i \leq t \,\} \qquad (2)$$

given $T_0, T_1, ..., T_{N-1}$.

As proved in Section 3.3, if $C$ is decodable at any time $t$, there must be at least $mn$ workers finishes computation. Consequently, we have

$$
\begin{aligned}
T &= \min\{\, t \geq 0 \mid C \text{ is decodable given results from all workers in } \mathcal{S}(t) \,\} \\
&= \min\{\, t \geq 0 \mid C \text{ is decodable given results from all workers in } \mathcal{S}(t) \text{ and } |\mathcal{S}(t)| \geq mn \,\} \\
&\geq \min\{\, t \geq 0 \mid |\mathcal{S}(t)| \geq mn \,\}. \qquad (3)
\end{aligned}
$$

On the other hand, we consider the latency of polynomial code, denoted by $T_{\text{poly}}$. Recall that for the polynomial code, the output $C$ is decodable if and only if at least $mn$ workers finishes computation, i.e., $|\mathcal{S}(t) \geq mn|$. We have

$$T_{\text{poly}} = \min\{\, t \geq 0 \mid |\mathcal{S}(t)| \geq mn \,\}. \qquad (4)$$

Hence, $T \geq T_{\text{poly}}$ always holds true, which proves Theorem 2.

## A.2  Proof of Theorem 3

Recall that in Section 3.3 we have proved that if the input matrices are sampled based on a certain distribution, then decoding the output $C$ requires that the entropy of the entire message received by the server is at least $rt \log_2 q$. Consequently, it takes at least $rt \log_2 q$ bits deliver such messages, which lower bounds the minimum communication load.

On the other hand, the polynomial code requires delivering $rt$ elements in $\mathbb{F}_q$ in total, which achieves this minimum communication load. Hence, the minimum communication load $L^*$ equals $rt \log_2 q$.

# Appendix B  Proof of Theorem 4

In this section, we formally describe a computation strategy, which achieves the recovery threshold stated in Theorem 4. Consider a distributed convolution problem with two input vectors

$$\boldsymbol{a} = [\boldsymbol{a}_0\ \boldsymbol{a}_1\ ...\ \boldsymbol{a}_{m-1}], \qquad \boldsymbol{b} = [\boldsymbol{b}_0\ \boldsymbol{b}_1\ ...\ \boldsymbol{b}_{n-1}], \qquad (5)$$

where the $\boldsymbol{a}_i$'s and $\boldsymbol{b}_i$'s are vectors of length $s$. We aim to compute $\boldsymbol{c} = \boldsymbol{a} * \boldsymbol{b}$ using $N$ workers. In previous literature [1], the computation strategies were designed so that the master can recover all intermediate values $\boldsymbol{a}_j * \boldsymbol{b}_k$'s. This is essentially the same computing framework used in the distributed matrix multiplication problem, so by naively applying the polynomial code (specifically the $(1, m)$-polynomial code using the notation in Definition 2), we can achieve the corresponding optimal recovery threshold in computing all $\boldsymbol{a}_j * \boldsymbol{b}_k$'s.

However, the master does not need to know each individual $\boldsymbol{a}_i * \boldsymbol{b}_j$ in order to recover the output $\boldsymbol{c}$. To customize the coding design so as to utilize this fact, we recall the general class of computation

strategies stated in Definition 2: Given design parameters $\alpha$ and $\beta$, the $(\alpha, \beta)$-polynomial code lets each worker $i$ store two vectors

$$\tilde{\boldsymbol{a}}_i = \sum_{j=0}^{m-1} \boldsymbol{a}_j x_i^{j\alpha}, \qquad \tilde{\boldsymbol{b}}_i = \sum_{j=0}^{n-1} \boldsymbol{b}_j x_i^{j\beta}, \tag{6}$$

where the $x_i$'s are $N$ distinct values assigned to the $N$ workers.

Recall that in the polynomial code designed for matrix multiplication, we picked values of $\alpha, \beta$ such that, in the local product, all coefficients $\boldsymbol{a}_j * \boldsymbol{b}_k$ are preserved as individual terms with distinct exponents on $x_i$. The fact that no two terms were combined leaves enough information to the master, so that it can decode any individual coefficient value from the intermediate result. Now that decoding all individual values is no longer required in the problem of convolution, we can design a new variation of the polynomial code to further improve recovery threshold, using design parameters $\alpha = \beta = 1$. In other words, each worker stores two vectors

$$\tilde{\boldsymbol{a}}_i = \sum_{j=0}^{m-1} \boldsymbol{a}_j x_i^j, \qquad \tilde{\boldsymbol{b}}_i = \sum_{j=0}^{n-1} \boldsymbol{b}_j x_i^j. \tag{7}$$

After computing the convolution product of the two locally stored vectors, each worker $i$ returns

$$\tilde{\boldsymbol{a}}_i * \tilde{\boldsymbol{b}}_i = \sum_{j=0}^{m-1} \sum_{k=0}^{n-1} \boldsymbol{a}_j * \boldsymbol{b}_k x_i^{j+k}, \tag{8}$$

which is essentially the value of the following degree $m + n - 2$ polynomial at point $x = x_i$.

$$h(x) = \sum_{j=0}^{m+n-2} \sum_{k=\max\{0, j-m+1\}}^{\min\{j,n-1\}} \boldsymbol{a}_{j-k} * \boldsymbol{b}_k x_i^j. \tag{9}$$

Using this design, instead of recovering all $\boldsymbol{a}_j * \boldsymbol{b}_k$'s, the server can only recover a subspace of their linear combinations. Interestingly, we can still recover $\boldsymbol{c}$ using these linear combinations, because it is easy to show that, if two values are combined in the same term of vector $\sum_{k=\max\{0, j-m+1\}}^{\min\{j,n-1\}} \boldsymbol{a}_{j-k} * \boldsymbol{b}_k$, then they are also combined in the same term of $\boldsymbol{c}$.

Consequently, after receiving the computing results from any $m + n - 1$ workers, the server can recover all the coefficients of $h(x)$, which allows recovering $\boldsymbol{c}$, which prove that this computation strategy achieves a recovery threshold of $m + n - 1$.

Similar to distributed matrix multiplication, this decoding process can be viewed as interpolating degree $m + n - 2$ polynomials of $\mathbb{F}_q$ for $s$ times. Consequently, the decoding complexity is $O(s(m+n)\log^2(m+n)\log\log(m+n))$, which is almost-linear to the input size $s(m+n)$.

*Remark* 1. Similar to distributed matrix multiplication, we can also extend this computation strategy to the scenario where the elements of input vectors are real or complex numbers, by quantizing all input values, embedding them into a finite field, and then directly applying our distributed convolution algorithm.

## Appendix C   Order-Wise Characterization of $K_{\text{conv}}$

Now we prove Theorem 5, which characterizes the optimum recovery threshold $K_{\text{conv}}$ within a factor of 2. The upper bound $K_{\text{conv}}^* \leq K_{\text{conv-poly}}$ directly follows from Theorem 4, hence we focus on proving the lower bound of $K_{\text{conv}}^*$. We first prove the following inequality.

$$K_{\text{conv}}^* \geq \max\{m, n\}. \tag{10}$$

Let $\boldsymbol{a}$ be any fixed non-zero vector, and $\boldsymbol{b}$ be sampled from $\mathbb{F}_q^{sn}$ uniformly at random. We can be easily show that the operation of convolving with $\boldsymbol{a}$ is invertible, and thus the entropy of $\boldsymbol{c} \triangleq \boldsymbol{a} * \boldsymbol{b}$ equals that of $\boldsymbol{b}$, which is $sn \log_2 q$. Note that each worker $i$ returns $\tilde{\boldsymbol{a}}_i * \tilde{\boldsymbol{b}}_i$, whose entropy is at most $H(\tilde{\boldsymbol{a}}_i) + H(\tilde{\boldsymbol{b}}_i) = s \log_2 q$. Using a cut-set bound around the master, we can show that at least $n$ results from the workers need to be collected, and thus we have $K^* \geq n$.

Similarly we have $K^* \geq m$, hence $K^* \geq \max\{m, n\}$. Thus we can show that the gap between the upper and lower bounds is no larger than 2: $K^* \geq \max\{m, n\} \geq \frac{m+n}{2} > \frac{m+n-1}{2} = \frac{K_{\text{conv-poly}}}{2}$.

# References

[1] S. Dutta, V. Cadambe, and P. Grover, "Coded convolution for parallel and distributed computing within a deadline," *arXiv preprint arXiv:1705.03875*, 2017.