[Reviews · NeurIPS 2017]

Reviewer 1



This paper extends the coded distributed computing of matrix multiplication proposed in Lee et al, 2016. The idea is to distribute the matrices by pre-multiplying them with the generator matrices of Reed Solomon code. Since Reed-Solomon codes are MDS codes, any K out of N distributed components are sufficient to recover the desired matrix multiplication. I am somewhat confused by this paper. Reed-Solomon codes ARE defined to be evaluations of polynomials over finite field (or any fields). So why the authors say that they define something new called polynomial code and then follow up by saying the decoding is just like Reed-Solomon code? Reed Solomon codes are MDS code (in fact, Lee et al also used Reed Solomon codes, they just called it MDS codes). The key idea of using codes for both matrices in matrix-matrix multiplication has appeared recently in Lee et al, 2017. I think there is a major problem from an application point of view in this paper that was not there for a simple matrix-vector multiplication of Lee et al of 2016. In case of matrix-vector multiplication A times x of Lee et al, 2016, the matrix A was fixed and x was variable. The matrix A was coded and distributed to the workers. x was uncoded. So the encoding is done only one time. It does indeed speed up the distributed computing. On the other hand, in this paper both the matrices A and B that are to be multiplied are being coded up. As a result when two matrices are given to multiply, a central node has to do a lot of polynomial evaluations and then distribute the evaluations among workers. The overhead for the central node seem to be way more than the actual computing task. Notably, this has to be done every time; as opposed to the one-time-encoding of Lee et al, 2016, or Dutta et al, 2016. Also surprising is the lack of experimental results. Most of the related literature is interesting because of the experimental superior performance they exhibit in real systems. This paper does not make any attempt in that direction. However since theoretically it is not that novel, and given the overhead in evaluating the polynomials, one really needs experimental validation of the methods. ======================== After the rebuttal: I did not initially see the making of the product MDS as a big deal (i.e. the condition that no two terms in the expression (16) have same exponent of x is not that difficult to satisfy). But on second thought, the simplicity should not be deterrent, given that the previous papers kept this problem open. There is one other point. What I understand is that each matrix is encoded using polynomials such that the product becomes codewords of Reed_Solomon codes. I do not necessarily agree with the term Polynomial code and the claim that a new class of code has been discovered. Finally, it is not easy to approximate matrix multiplications over reals with finite field multiplication. I don't know why that is being seen as straight-forward quantization.

Reviewer 2



The paper presents optimal error correcting based technique to speed up matrix-matrix multiplication. The authors follow a new line of work that uses ideas from coding theory to alleviate the effect of straggler (eg slower than average) nodes in distributed machine learning. The main idea of coding in the context of distributed matrix multiplication is the following (a simple 3 worker example): Let A = [A1;A2], and a setup of 3 workers and a master. If a worker is computing A1*B, a second worker is computing A2*B, and a third is computing a (A1+A2)*B , then from any 2 workers a master node could recover A*B, hence not having to wait for all 3 of the nodes. This idea can both provably and in practice improve the performance of distributed algorithms for simple linear problems like matrix multiplication, or linear regression. In this work, the authors present information-theoretically optimal codes for matrix-matrix multiplication, with respect to the number of nodes the master has to wait to recover the full matrix multiply. The novelty in the presented work is that coding is performed on both A and B matrices, and most importantly it is done in such a way that if the master can recover the A*B result from as few nodes as information theoretically possible: eg if each worker can store a 1/r fraction of A and B, then by waiting for only r^2 nodes the master can recover the results. The novel coding technique used to achieve the above optimal result is called polynomial codes, and comes with several benefits, such as fast decoding complexity due to its Reed-Solomon structure. The authors couple their “achievability” result with a lower bound, essentially showing that polynomial codes are optimal with respect to the number of workers the master needs to wait for before recovering the full matrix multiplication result. Overall, the paper is well written, and the theoretical contribution is strong and of interest to a new and emerging research direction on the intersection of Coding Theory and Distributed Machine Learning. My only question that is not addressed by the paper, is how well does the presented theoretical scheme work in practice. It would be useful to see some experiments that validate the theoretical results.

Reviewer 3



Summary: The authors consider the problem of efficient distributed multiplication of two matrices when there are many workers and some of them could be straggling. One simple metric the authors consider is optimizing the minimum number of workers that need to finish before they can recover the full matrix product. Suppose matrices that are being multiplied are A and B. Suppose there are N workers. Suppose A and B are divided into m and n parts each respectively. The previous best bounds suggested that \Theta(sqrt(N)) workers need to finish before recovery. In this work, authors propose a construction called polynomial code by which one needs only mn workers to finish which is independent of N. Further it is also information theoretically optimal. They also show that a similar construction can work for convolution which requires only m+n-1 workers to finish. The authors show that in the multiplication case, because it is optimal with respect to this measure, it is also optimal when one takes into account any computational latency model for delays in the workers and also in terms of communication load. The key idea is for every worker to evaluate a polynomial in the A-parts with degree profile increasing in steps of 1 and evaluate another polynomial in the B-parts with degree profile increasing in steps of m. The worker multiplies both polynomial evaluations of A-parts and B-parts to master node. It turns out that all mn product parts of A and B, are coefficients of a polynomial of degree mn. If mn workers return evaluations, it is evaluated in mn points which helps the master to recover using polynomial interpolation or Reed Solomon decoding. Strengths: a) The idea of using polynomial evaluations for both A-parts and B-parts is the key idea and the authors propose a simple solution for distributed multiplication and convolution. Weaknesses: My only main concern with the paper is as follows: a) The main weakness is that it lacks some real experiments with real matrices. The theory is for finite fields. Authors mention that in Remark 6 (also elsewhere) that one can quantize reals in finite number of bits and then actually consider the quantized bits as elements of a finite field and perform such multiplication using polynomial codes. However there are hidden issues of dynamic range of real entries involved, size of the field required and presumably polynomial interpolation algorithms become difficult to implement in with large field size owing to quantization. If such things are taken into account, then clearly more discussion is needed. Very classical Reed Solomon codes in practice are implemented with field size of 256 (8 bit symbol). Its not clear if there are issues with scaling the field size beyond this. A practical experiment (with synthetic examples) could have ratified this point. This is because I can only see the most practical use case in ML system to be a problem with reals.